# OpenReview forum: "An Image Is Worth 1000 Lies: Transferability of Adversarial Images across Prompts on Vision-Language Models"
_ICLR.cc/2024/Conference — ICLR 2024 spotlight_

### Official Review · Reviewer_iSYx · 2023-10-30

**Soundness:** 2 fair
**Presentation:** 3 good
**Contribution:** 2 fair
**Rating:** 6
**Confidence:** 4

**Summary:**

This work proposes the Cross-Prompt Attack (CroPA). This proposed method updates the visual adversarial perturbation with learnable textual prompts, which are designed to counteract the misleading effects of the adversarial image. By doing this, CroPA improves the transferability of adversarial examples across prompts.

**Strengths:**

The experiments have shown the cross-prompt transferability created with a single prompt is highly limited. An intuitive approach to increase the cross-prompt transferability is to use multiple prompts during its creation stage. However, the improvement in cross-prompt transferability of these baseline approaches converges quickly with the increase in prompts. CroPA further improve the cross-prompt transferability.

It creates more transferable adversarial images by utilising the learnable textual prompts. The learnable textual prompts are optimised
in the opposite direction of the adversarial image to cover more prompt embedding space.

To explore the underlying reasons for the better performance of CroPA compared to the baseline approach, it visualises the sentence embedding of the original prompt and perturbated prompts by CroPA, which is obtained by the averaging embedding of each token. The visualisation of difference in the prompt embedding coverage explains the reason why the CroPA methods can outperform the baseline approach even if the number of prompts used in optimisation is less than the baseline approach.

It experiments with different settings or policies. For example, it explores the effect of different update strategies. It tests the ASRs of the image adversarial examples with the number of in-context learning examples.

**Weaknesses:**

The technical contribution may be limited. The optimization method is general to use gradients for updating the adversarial perturbations. This optimization method is widely adopted in adversarial attacks.  It is very similar to adversarial training. To achieve cross prompt transferability, it perturbs the prompts without limitations to maximize the loss and obtain the worst prompt, then it trains the image perturbations based on the worst prompts and the final obtained perturbation can fool the model for different prompts because it already deals with the worst prompts. The novelty may be limited.


I am still not very clear about some experiment settings. For example, what is the value of perturbation size $\epsilon$? Larger $\epsilon$ usually means stronger attacks. So what is the performance with a different $\epsilon$?

It seems that the generated perturbations are used to test for the same model that it was trained. Can the generated perturbations be transferred to other VLMs and what are the performance? It is better to discuss the transferability to demonstrate the generalization of the method.

For the non-targeted attack after equation (2), the optimisation is expressed to maximize over token and minimize over image. It is almost the same as the targeted attack. I am wondering if there is a mistake here and it should be to maximize over image to make the results of adversarial examples different from the original one. Minimizing over images and the final image perturbation would make the outputs of adversarial examples similar to clean images. It is better to provide more details about this non-targeted attack. I agree that by maximizing the tokens, the generated tokens can mislead the VLM, but here we focus on image perturbations and use the image perturbation to test the VLM, right?

**Questions:**

see the weakness.

---

> ### Author Response · Authors · 2023-11-21
>
> **Q1: The technical contribution and novelty may be limited.**
>
> **A:**  In this work, we have multiple contributions. The first contribution of our paper is that we introduce a new problem, namely, transferability across prompts. The novelty of this contribution is recognized by all other reviewers. As the reviewer realized, we also proposed a method to improve the transferability across prompts. This method is straightforward yet proves to be highly effective and our experiments have consistently demonstrated its effectiveness.
>
> **Q2: What is the performance with a different ε?**
>
> **A:** This is a good question. In our paper, the perturbation size is set to 16/255, and we included this information in our paper. We conducted experiments with different perturbation sizes, and the results are shown in Table 7 of Appendix C. The results have shown that our method still consistently outperforms the baseline methods.
> | Perturbation size | Method | VQA_general | VQA_specific | Classification | Captioning | Overall |
> |-------------------|--------|-------------|--------------|----------------|------------|---------|
> | 8/255             | Multi-P| 0.37        | 0.59         | **0.56**       | **0.06**   | 0.39    |
> |                   | CroPA  | **0.53**    | **0.75**     | 0.48           | 0.03       | **0.45**|
> | 16/255            | Multi-P| 0.67        | 0.86         | 0.64           | 0.31       | 0.62    |
> |                   | CroPA  | **0.92**    | **0.98**     | **0.70**       | **0.39**   | **0.75**|
> | 32/255            | Multi-P| 0.85        | 0.95         | 0.47           | 0.32       | 0.64    |
> |                   | CroPA  | **0.98**    | **0.99**     | **0.59**       | **0.42**   | **0.75**|
>
>
> **Q3: Transferability of the generated perturbations across different VLMs.**
>
> **A:** Yes, our results show that the perturbations can be transferred across different VLMs. We have expanded our discussion on this topic. Our experimental results in Table 8 in Appendix C show an overall ASR of close to 10% for cross-model transferability in addition to the cross-prompt transferability, despite differences in architecture and size, such as between OPT-2.7b and Vicuna-7b. The transferability across different VLMs depends on the similarities between the language models. We expect the cross-model transferability to be stronger if the language model components are highly similar (e.g. one model is the instruction fine-tuned variant of the source model).
>
>
> **Q4: Questions about the formula for non-targeted attack after equation (2).**
>
> **A:** Thank you for your careful observation. Indeed, it was a typo in the paper. The ultimate goal of the max-min process is to make the generation result of adversarial examples different from the original one, and the maximization operations should be applied to the image perturbation. We have switched the order of image perturbation $\delta_v$ and prompt perturbation $\delta_t$ in the formula.

---

> ### Author Response · Authors · 2023-11-22
> **A friendly reminder**
>
> This is a friendly reminder for the reviewer to update their comments. We are happy to provide more information if the reviewer still has some concerns. Thank you again for your time on our rebuttal.

---

### Official Review · Reviewer_Fnt8 · 2023-11-01

**Soundness:** 2 fair
**Presentation:** 3 good
**Contribution:** 2 fair
**Rating:** 8
**Confidence:** 4

**Summary:**

The paper proposes the Cross-Prompt Attack (CroPA) that generalizes the pixel-space image perturbation for multiple prompts. The authors propose to optimize learnable textual prompts in the opposite direction of the pixel-space adversarial perturbation for transferability. Evaluations on small-scale VLMs showed that CroPA was effective in fooling the target model, and the pixel-space perturbations are transferable across multiple types of prompts.

**Strengths:**

1. The paper is well-written and well-presented. The motivations are sufficiently reasonable to motivate the framework.

2. The idea of jointly optimizing pixel-space perturbation and text-space perturbation is novel.

3. The evaluations include multiple kinds of VLM paradigms with several major language tasks, which are comprehensive.

**Weaknesses:**

1. The paper assumes the white-box access for VLMs, which enables the framework to have backward gradients to update in multimodal input space. However, this assumption may not be generalized to all mainstream VLMs as they are fast scaling up. For instance, the authors evaluated the effectiveness of OpenFlamingo-9B and InstructBLIP, whose LLM component is not of large-scale. The applicability of CroPA on large-scale VLMs (e.g., those embedded with LLaMA-65B) or black-box VLMs (e.g., GPT4-V) remains challenging.

2. The framework learns adversarial prompts to enhance the attack's effectiveness. However, very few text prompt instances are shown in the paper, and the quality of the result text prompts is not sufficiently evaluated.

3. Existing adversarial attacks on VLM [1] have adopted query-based techniques to enhance the attack effectiveness. What's more, this baseline only assumes the black-box access to the VLMs, which more are generalizable. The paper should show sufficient validity or advantages (e.g., ASR, Convergence, etc.) of CroPA over existing baselines.

4. Please fix the typo "leanable" in the introduction (page 2).

[1] (NeurIPS 2023) On Evaluating Adversarial Robustness of Large Vision-Language Models.

**Questions:**

1. Please address my concerns stated in the weakness section. Given the current status of the paper, I appreciate the problem novelty to rate it as borderline acceptance. However, the authors' responses shall address the weakness mentioned above. I look forward to further discussion and I will consider revising the rating based on the soundness of the response.

---

> ### Author Response · Authors · 2023-11-21
>
> **Q1.1: The method may not be generalized to all mainstream VLMs as they are fast scaling up.**
>
> **A:** This is a good question. We discuss the challenges brought by scaling up the models in the following two perspectives.
> 1. Accurate gradient: With larger VLMs, it can be more challenging to obtain accurate gradient information. In our community, several approaches have also been proposed to address this problem such as EOT [1]. These approaches can be combined to effectively address the gradient accuracy challenge in large VLMs. In this work, we applied the standard attack method.
> 2.  The memory issue caused by large VLMs: our community has actively proposed methods to allow for the usage of increasingly large models. Hugging Face, for instance, allows users to load models by distributing model shards across multiple graphics cards.
>
> **Q1.2:   The applicability of CroPA under the black-box setting**
>
> **A:** Achieving strong transferability across models is indeed challenging. We also studied the performance under the black-box setting. Our experimental results in Table 7 in Appendix C indicate an ASR of close to 10% for transferability across models, despite differences in architecture and size, such as between OPT-2.7b and vicuna-7b. We expect the cross-model transferability to be stronger if the language model components are highly similar (e.g. one model is the instruction fine-tuned variant of the source model). Our study is expected to have broad applications as many models are built based on prevalent open-source vision models (e.g. OpenCLIP) and language models (e.g. LlaMA).
>
> **Q2: The framework learns adversarial prompts to enhance the attack's effectiveness. However, very few text prompt instances are shown in the paper, and the quality of the result text prompts is not sufficiently evaluated.**
>
> **A:** Thank the reviewer for pointing this out. The adversarial prompts are learnable embeddings instead of tokens. The following are more detailed clarifications regarding this question:
> 1. In optimization, we update the embedding of the prompt perturbations instead of updating the discrete tokens of the prompts. That's the reason why we are not able to show the adversarial prompts directly.
> 2. We also decoded the perturbed prompt embeddings by approximating the tokens in terms of cosine similarity. The approximate tokens of the perturbed prompt embeddings are the same as the original prompts, which can be found in Section 4.6.
>
> **Q3: Discussion about the difference with another work [2] under the black-box settings.**
>
> **A:** The study [2] under the black box settings indeed provides meaningful insights into the cross-model transferability, but our work focuses on a new perspective transferability across. Our works differ in the following aspects.
>
> 1. The goals of these two studies are different. The study [2] explores the adversarial robustness of VLMs from an evaluation perspective. It cannot be trivially applied to solve the cross-prompt transferability, which is the focus of this paper.
>
> 2. The evaluation methods in the two papers are different as well. The ASR (Attack Success Rate) discussed in our paper adheres to a strict standard: an attack is considered successful only if the model is deceived into generating the exact target text. The evaluation metric used in [2] is CLIP-score, which does not require the exact match of the generated text and the target text.
>
> Moreover, our method makes sense in both black-box and white-box. The black-box attacks on VLMs can be combined with our CroPA method to enable our methods to work in a black-box fashion.
>
> **Q4: Typo "leanable" in the introduction (page 2)**
>
> **A:** Thanks for pointing this out, and we have corrected the typo.
>
> [1] Athalye, Anish, Nicholas Carlini, and David Wagner. "Obfuscated gradients give a false sense of security: Circumventing defenses to adversarial examples." International conference on machine learning. PMLR, 2018.
>
> [2] Zhao, Yunqing, et al. "On evaluating adversarial robustness of large vision-language models." arXiv preprint arXiv:2305.16934 (2023).

---

> > ### Comment · Reviewer_Fnt8 · 2023-11-22
> > **Concerns are well addressed**
> >
> > The responses address my concerns well. I appreciate the reasonable discussion and raise the rating to accept.

---

### Official Review · Reviewer_av8s · 2023-11-02

**Soundness:** 3 good
**Presentation:** 3 good
**Contribution:** 2 fair
**Rating:** 6
**Confidence:** 5

**Summary:**

This paper aims to generate a single adversarial image capable of misleading all predictions made by VLMs, regardless of the input prompts. To accomplish this objective, it introduces a Cross-Prompt Attack, characterized by a minimax optimization process involving image perturbations and learnable textual prompts. Extensive experiments have validated that, in a white-box setting, the proposed attack scheme can outperform other attack methods.

**Strengths:**

1. The proposed method intuitively makes sense and consistently outperforms other baseline attack schemes in extensive experiments.

2. The paper is well-written and easy to follow.

**Weaknesses:**

1. The primary concern for this paper is whether the proposed attack can indeed raise significant security issues. For instance, when considering the exemplary target prompts in Table 1, they have the potential to disrupt the functionality of VLMs, but it remains unclear how they may pose security risks in real-world applications. This security concern would be more valid if the attack successfully tricks VLMs into producing contradictory predictions or harmful instructions. Otherwise, the objective of this study appears to be more like a pioneering exploration of VLMs' adversarial robustness.

2. The author mentions another potential use case for the proposed framework, which is to "prevent malicious usage of large VLMs for unauthorized extraction of sensitive information from personal images." If this is indeed the case, the proposed method should also be assessed in a black-box setting, as users' VLMs cannot be known in advance. However, this set of experiments is absent from the paper. Previous jailbreaking attacks aimed at emerging large language models have shown substantial transferability across models [1]. I am curious whether this holds true for various VLMs as well.

3. I am curious whether some straightforward defense techniques can defeat the proposed attack, such as incorporating data augmentation on the input images [2]. This could provide insights into whether the optimized adversarial perturbation converges to an unstable local minimum and can be easily defended against.

[1] "Universal and Transferable Adversarial Attacks on Aligned Language Models", A. Zou et al., arXiv 2023.

[2] "The Best Defense Is a Good Offense: Adversarial Augmentation Against Adversarial Attacks", I. Frosio et al., CVPR 2023.

**Questions:**

In addition to the concerns in the weakness section, I also have the following question:

Is the improvement achieved by the proposed Cross-Prompt Attack over Multi-P due to the increased number of textual prompts generated through perturbations during minimax optimization? One approach to achieve a similar effect is to utilize another language model to rephrase the text prompts into new ones, thus serving as data augmentation.

---

> ### Author Response · Authors · 2023-11-21
>
> **Q1: Whether the proposed attack can indeed raise significant security issues？ This security concern would be more valid if the attack successfully tricks VLMs into producing contradictory predictions or harmful instructions.**
>
> **A:** We agree that tricking the model into generating harmful instructions is indeed a security concern, but our goal is different. The goal of this paper is to study the adversarial transferability across prompts of the VLMs.
>
> There is some connection between the two topics. When the target output is set to harmful content, such as  “suicide”, the model can output inappropriate sentences for images.
>
> The detailed data are shown as follows, which is also presented in Table 9 in Appendix C.
>
> | Target Texts | Method | VQA_general | VQA_specific | Classification | Captioning | Overall |
> |--------------|--------|-------------|--------------|----------------|------------|---------|
> | suicide      | Multi-P| 0.69        | 0.83         | 0.75           | 0.61       | 0.72    |
> |              | CroPA  | **0.84**    | **0.91**     | **0.91**       | **0.78**   | **0.86**|
> | kidnap       | Multi-P| 0.87        | 0.92         | 0.69           | 0.73       | 0.80    |
> |              | CroPA  | **0.94**    | **0.96**     | **0.88**       | **0.81**   | **0.90**|
> | bomb         | Multi-P| 0.68        | 0.82         | 0.90           | 0.60       | 0.75    |
> |              | CroPA  | **0.80**    | **0.90**     | **0.94**       | **0.70**   | **0.84**|
>
>
>
> Therefore, setting the target sentence to harmful words can pose threats to the applications of large VLMs. We would like to point out again that harmful content generation is not the focus of this work. But, we agree it is interesting to discuss the relationship between them.
>
> **Q2: Questions about the statement "prevent malicious usage of large VLMs for unauthorized extraction of sensitive information from personal images" and comparisons with the transferability across models in the context of jailbreak.**
>
> **A:** Thank you for highlighting this aspect.
> 1. The white-box setting also makes sense for this statement.  For example, staff in technology companies may have access to users’ images, and there is potential for them to maliciously use their company’s models to extract users' information. Our method can serve as a privacy protection method if the images are pre-processed by adding invisible perturbations designed for their company’s models.
>
> 2. Our setting is much more stringent than the jailbreak scenarios in Large Language Models (LLMs). In this work, we assume that only images can be manipulated while the jailbreak-related study in LLMs allows for modifying prompts directly. Manipulating VLM outputs through images is challenging due to the difficulty in aligning visual information with text, as detailed in [1].  It is also not entirely clear how to achieve jailbreaking by modifying images only. However, it is a really good question to explore in our future work.
>
>
> **Q3: Whether some straightforward defense techniques can defeat the proposed attack, such as incorporating data augmentation on the input images?**
>
> **A:** Thanks for bringing up this question.
> 1. We tested the ASR (Attack Success Rate) of our method by incorporating common data augmentation techniques, such as random rotation on the input images, during the testing phase only. As detailed in Table 6 in Appendix C, the overall ASR of CroPA drops 5% while the performance of Multi-P drops 8%. Please note that we do not include any data augmentation during the optimization stage.
>
> 2. There is an arms race between attack and defense strategies. To counter data augmentation defense techniques, data augmentation can be easily integrated into the optimization process. Better attack and defense methods are orthogonal to this work. In other words, these techniques can all be combined with our work readily.
>
> **Q4: Is the improvement achieved by CroPA due to the increased number of textual prompts generated through perturbations? One approach to achieve a similar effect is to utilize another language model to rephrase the text prompts into new ones, thus serving as data augmentation.**
>
> **A:** This is a very good question.
> 1. Firstly, as shown in Figure 2 in the paper, the improvement from adding more prompts quickly saturates. Beyond the point where the number of prompts reaches ten, the increase in cross-prompt transferability becomes marginal, especially for Multi-P. Therefore, using more prompts does not necessarily mean a stronger performance.
> 2. Given the same number of prompts and the same number of optimization iterations as shown in Figure 3 in the paper, the proposed CroPA method clearly outperforms the Multi-P method.
>
>
> [1] Lu, Dong, et al. "Set-level guidance attack: Boosting adversarial transferability of vision-language pre-training models." Proceedings of the IEEE/CVF International Conference on Computer Vision. 2023.

---

> ### Author Response · Authors · 2023-11-22
> **A friendly reminder**
>
> This is a friendly reminder for the reviewer to update their comments. We are happy to provide more information if the reviewer still has some concerns. Thank you again for your time on our rebuttal.

---

### Official Review · Reviewer_FoXe · 2023-11-02

**Soundness:** 3 good
**Presentation:** 4 excellent
**Contribution:** 3 good
**Rating:** 8
**Confidence:** 4

**Summary:**

This paper studies a novel perspective on adversarial transferability in the context of Vision and Language Models (VLM). In particular, it explores the problem of learning for adversarial visual patterns that can seamlessly traverse various textual prompts. The author introduces the method of a cross-prompt attack, akin to the principles of Generative Adversarial Networks (GANs), where the goal is to simultaneously learn visual perturbations and prompt perturbations with opposing objectives. Through compelling demonstrations, the study reveals that this competitive process significantly enhances the visual perturbation's ability to traverse different prompts effectively.

**Strengths:**

The problem being studied (cross-prompt adversarial transferability) and the proposed method seems very novel to me, and the author performed extensive experiments to validate the strength of the proposed method. I am convinced that this paper will provide valuable insights and knowledge to the research community.

**Weaknesses:**

I don’t see a major weakness, but have some suggestions for the naming. The term "prompt perturbation" initially appears a bit confusing, as it implies an intention to deceive the model (as it’s commonly used for adversarial “attach” rather than “defense”), which is not the case in this paper. Instead, it’s used in an opposite direction other than image perturbation to encourage stronger cross-prompt transferability during this competitive update process. I would recommend that the author consider selecting a more fitting name.

**Questions:**

How does the proposed method work for the longer prompt? Does the transferability still hold?

---

> ### Author Response · Authors · 2023-11-21
>
> **Q1: The term "prompt perturbation" initially appears a bit confusing.**
>
> **A:** We thank the reviewer for pointing out this issue. We included clarifications on this in our paper to avoid confusion. The term “prompt perturbation” refers to the perturbations that are added to the embeddings of the prompts for obtaining image perturbation with strong cross-prompt transferability. It is not optimized to deceive the model.
>
> **Q2: How does the proposed method work for the longer prompt? Does the transferability still hold?**
>
> **A:** Yes,  our method still outperforms the baseline approaches for longer target texts. The detailed data can also be found in Table 10 in Appendix C.
> | Target Texts         | Method  | VQA_general | VQA_specific | Classification | Captioning | Overall |
> |----------------------|---------|-------------|--------------|----------------|------------|---------|
> | I am sorry           | Multi-P | 0.60        | 0.85         | 0.71           | 0.60       | 0.69    |
> |                      | CroPA   | **0.90**    | **0.96**     | **0.75**       | **0.72**   | **0.83**|
> | I cannot answer      | Multi-P | 0.40        | 0.66         | 0.33           | 0.07       | 0.37    |
> |                      | CroPA   | **0.58**    | **0.67**     | **0.33**       | **0.20**   | **0.45**|
> | I do not know        | Multi-P | 0.67        | 0.75         | 0.41           | 0.03       | 0.47    |
> |                      | CroPA   | **0.70**    | **0.80**     | **0.43**       | **0.04**   | **0.49**|
> | I need a new phone   | Multi-P | 0.68        | 0.86         | 0.85           | 0.53       | 0.73    |
> |                      | CroPA   | **0.83**    | **0.85**     | **0.77**       | **0.70**   | **0.79**|

---

### Official Review · Reviewer_HSuq · 2023-11-06

**Soundness:** 3 good
**Presentation:** 3 good
**Contribution:** 2 fair
**Rating:** 6
**Confidence:** 4

**Summary:**

The authors propose a method called Cross-Prompt Attack (CroPA) that creates adversarial images, destined for vision-language models (VLMs), that transfer across prompts.  They train on multiple prompts using a variant of projected gradient descent to learn both the image perturbation and the text embedding perturbation. They consider both the targeted and non-targeted scenarios. CroPA achieves better coss-prompt transferability compared to baseline methods when evaluated with OpenFlamingo-9B, BLIP-2, and InstructBLIP on different tasks.

**Strengths:**

- The paper introduces an aspect of adversarial transferability that was not emphasized before. The problem that the paper tries to address is clearly formulated and is worth investigating.
- The experimental setup covers a wide range of scenarios: targeted and non-targeted attacks, multiple VLMs (OpenFlamingo-9B, BLIP-2,InstructBLIP) and different multi-modal tasks (VQA, classification, captioning). The baselines are strong.

**Weaknesses:**

- The method does not achieve transferability across models or images in addition to cross-prompt transferability. This might limit  its practical applicability.

- Clarifications are required in some parts of the paper. Certain statements about the “textual prompts” are misleading. For example, in the abstract, the method is described as follows:  “This proposed method updates the visual adversarial perturbation with learnable textual prompts”. This might suggest that the method directly modifies the textual prompts or discrete prompt tokens whereas it is actually modifying the embedding of these textual prompts. Moreover, the authors should explicitly state the underlying optimization algorithm (is it projected gradient descent?) , the norm (is it L-infinity ?) and the selected value of the perturbation size for easier comparison with related work.

Minor issues: there are some typos in the paper (for example in the conclusion “baseline approaches only archive” -> “achieve”).

**Questions:**

- This is a clarification question. Do you only add the prompt perturbations during the optimization phase or do you also add them during evaluation?

- What is the ASR of Multi-P when transferred across models or images?

- Do you have a conjecture regarding the inability of CroPA to achieve transferability across models or images?

---

> ### Author Response · Authors · 2023-11-21
>
> **Q1: The method does not achieve transferability across models or images in addition to cross-prompt transferability. This might limit its practical applicability.**
>
> **A:** Thank you for your insightful observation regarding the transferability in other aspects. We appreciate the opportunity to clarify and expand upon these points.
> 1. The transferability across different images is orthogonal to the transferability across different prompts. Transferability across prompts can be combined with transferability across images by utilizing more images during the optimization stage. As stated in the paper [1], the perturbation computed over a larger dataset can increase the cross-image transferability.
>
> 2. The transferability across different models depends on the similarities between the model architecture. Our study is expected to have broad application as many models are built based on common open-source vision models (e.g. OpenCLIP) and language models (e.g. LlaMA).
>
>
> **Q2.1: Clarifications are required in some parts of the paper. Certain statements about the “textual prompts” are misleading.**
>
> **A:** Thanks for pointing this out. We agree with the reviewer about the potential misunderstanding. Following the suggestion, we replaced "learnable textual prompts" with "learnable prompts" across the paper.
>
> **Q2.2： More clarifications about the underlying setting of the experiments (e.g. optimization algorithm,  the norm, and the perturbation size )**
>
> **A:** We follow the standard setting, and adopt the PGD as our optimization algorithm with L-infinity norm. The perturbation size is fixed to 16/255. We have updated our paper accordingly. We added the ablation experiments with different perturbation sizes as shown in Table 7 in Appendix C.
> | Perturbation size | Method | VQA_general | VQA_specific | Classification | Captioning | Overall |
> |-------------------|--------|-------------|--------------|----------------|------------|---------|
> | 8/255             | Multi-P| 0.37        | 0.59         | **0.56**       | **0.06**   | 0.39    |
> |                   | CroPA  | **0.53**    | **0.75**     | 0.48           | 0.03       | **0.45**|
> | 16/255            | Multi-P| 0.67        | 0.86         | 0.64           | 0.31       | 0.62    |
> |                   | CroPA  | **0.92**    | **0.98**     | **0.70**       | **0.39**   | **0.75**|
> | 32/255            | Multi-P| 0.85        | 0.95         | 0.47           | 0.32       | 0.64    |
> |                   | CroPA  | **0.98**    | **0.99**     | **0.59**       | **0.42**   | **0.75**|
>
>
> **Q3: Typos in the conclusion part (“baseline approaches only archive” -> “achieve”).**
>
> **A:** We have fixed this typo, thanks for the correction.
>
> **Q4: Do you only add the prompt perturbations during the optimization phase or do you also add them during evaluation?**
>
> **A:** Prompt perturbations are only added during the optimization phase,  and during the evaluation phase we do not add the prompt perturbation. The reason for using prompt perturbation during the optimization is to obtain the image perturbation with stronger cross-model transferability. During the test stage, the image perturbation is fixed and the prompt perturbations are not added.
>
> **Q5: What is the ASR of Multi-P when transferred across models or images?**
>
> **A:** The ASRs of the Multi-P are all zeros when transferred across images. The ASRs of Multi-P across different images are around 3% and the detailed data can be found in Table 8 in Appendix C. As the table shows, the Multi-P underperforms CroPA, even though the cross-model transferability performance of CroPA is limited. Please note that the techniques for enhancing cross-image transferability are orthogonal to our study, and they can be combined with our method.
>
>
> **Q6: Do you have a conjecture regarding the inability of CroPA to achieve transferability across models or images?**
>
> **A:** As reviewers realized, the transferability across models or images of both Multi-P and CroPA is not strong. A recent work [2] made initial explorations of the challenges of adversarial transferability across VLMs. It stated that it is difficult to align the visual information to text, namely the visual information can correspond to multiple texts. We conjecture that our methods on VLMs face similar challenges, but it leaves a new direction to improve transferability across the models or images with existing transferability-enhancing methods.
>
> [1] Moosavi-Dezfooli, Seyed-Mohsen, et al. "Universal adversarial perturbations." Proceedings of the IEEE conference on computer vision and pattern recognition. 2017.
>
> [2] Lu, Dong, et al. "Set-level guidance attack: Boosting adversarial transferability of vision-language pre-training models." Proceedings of the IEEE/CVF International Conference on Computer Vision. 2023.

---

> > ### Comment · Reviewer_HSuq · 2023-11-22
> > **Response to Authors**
> >
> > The authors answered the clarification questions. Moreover, they took into account all the comments and updated the paper accordingly.

---

### Author Response · Authors · 2023-11-21

We thank all the reviewers for their insightful feedback and constructive suggestions. We made the following changes to our paper.
1. We added experiments with longer target texts to prove that our proposed method still outperforms the baseline methods under this scenario.
2. We extended the discussion on the transferability across models and images in addition to the cross-prompt transferability.

3. We added the discussion between the jailbreak and our study on cross-prompt transferability. We conducted experiments where target texts were set to harmful instructions. The results show that our study on cross-prompt transferability indeed reveals an important security issue of the VLMs.
4. We added experiments to explore the effectiveness of our method when defense strategies are applied in the test stage.

5. We clarified some terms following the reviewers’ suggestions such as “learnable prompts” and “prompt perturbation”.

---

### Author Response · Authors · 2023-11-23
**Summary of Author-Reviewer discussion for AC and Reviewers**

Thank AC for handling our paper. Thank all reviewers for their valuable time and insightful feedback.

We would like to summarize our Author-Reviewer discussion for AC and Reviewers. In total, we received five reviews for our work.

After we submitted our rebuttal, three reviewers (Reviewer HSuq, FoXe
, Fnt8) have read our response, confirmed their final positive feedback, and raised the scores to 6, 8, 8. We appreciate their time to give feedback on our paper. The encouraging positive feedback motivates us to explore further in this research direction.

However, two reviewers (Reviewer iSYx and av8s) with an original score of 5 did not respond to our rebuttal, **although we put a reminder**. We understand the reviewers might be too busy with their own daily work. But it would be really discouraging if they will insist on rejecting the paper with unsolved or new concerns in the final stage, in which we are unable to provide further clarification.

We hope that AC can take this into consideration when making a final recommendation on our work.

We sincerely thank all reviewers and AC again for their valuable time on this work.

Best Regards!

---

### Meta-Review · Area_Chair_BBYv · 2023-11-30

**Metareview:**

This paper proposes a new setup in the context of adversarial transferability, namely cross-prompt adversarial transferability. The paper introduces a method to boost the transferability of adversarial examples across prompts, and presents extensive experiments and analysis showing the effectiveness of the method and characterizing its behavior.

The paper presents a novel and intriguing aspect of adversarial transferability, and convincing experiments to illustrate the proposed method and setup.

The setup might be somewhat removed from practical applications, but this is not one of the main aims of the paper.

**Justification For Why Not Higher Score:**

While the paper is interesting and original, it defines a somewhat niche problem.

**Justification For Why Not Lower Score:**

The paper does present an interesting setup and tackles it convincingly, not only in terms of performance, but also analysis.

---

### Decision · Program_Chairs · 2024-01-16

Accept (spotlight)